# Symbiotic Tuning: A Simple Approach for Enhancing Task Performance of Side-Tuning

## Abstract

The reduction of the computational and memory overhead associated with fine-tuning large language models remains a significant challenge for current research in natural language processing. Achieving an optimal balance between task performance, adaptability, and low VRAM requirement often presents a complex trade-off. Parameter-efficient fine-tuning (PEFT) methods, such as LoRA, have gained attention for their ability to reduce the number of trainable parameters while preserving task performance. However, they have not yet achieved a notable reduction in VRAM usage, which is still predominantly consumed by model weights and activations during backpropagation. In contrast, Ladder Side-Tuning (LST) has been proposed as an alternative that effectively reduces VRAM usage by freezing the backbone language model (BLM) and training only lightweight side networks. Nevertheless, this reduction in memory usage often results in a decline in performance, as LST typically exhibits inferior performance compared to PEFT methods on the same BLM. To address these limitations, we propose Symbiotic Tuning (SymTune), a novel approach that extracts intermediate outputs from the BLM and integrates symbiotic modules to enhance feature processing capabilities. This method avoids a direct trade-off between performance and VRAM efficiency, offering two key advantages: 1) robust performance across a wide range of natural language tasks, and 2) reduced VRAM consumption through an improved side-tuning architecture. The experimental results demonstrate that SymTune provides a scalable and memory-efficient solution for fine-tuning language models.

## 1 Introduction

*The VRAM bottleneck of traditional PEFT*: During model training, VRAM consumption is primarily dominated by three components: model weights, gradients, and activations. Parameter-Efficient Fine-Tuning (PEFT) methods, such as LoRA, have proven effective in transfer learning for downstream tasks by significantly reducing the number of trainable parameters, directly decreasing the memory required for storing gradients. However, this reduction only addresses the gradient-related VRAM usage. The memory demands for the other two components—model weights and activations—remain high and can still accumulate to hundreds of gigabytes. Consequently, while reducing the trainable parameter count alleviates some of the memory burden, it is insufficient to fully mitigate the VRAM bottleneck in large-scale models. To address this challenge, more comprehensive solutions targeting activations and model weights are necessary.

Ladder Side-Tuning (LST) (Sung et al., 2022) offers a promising solution to address the memory bottleneck in fine-tuning large models. By drastically reducing the number of trainable parameters, LST frees up a significant portion of the VRAM typically consumed by activations during training. Additionally, LST can further optimize memory efficiency through quantization techniques (Zhang et al., 2024), which compress the model weights by representing them with lower-precision data types. This approach makes side-tuning particularly useful for fine-tuning in environments where VRAM is a limiting factor.

*The Performance Limitation of Side-tuning*: Side-tuning methods freeze the backbone language model (BLM) and apply gradients only to the side network. While this approach is fast and memory-efficient, it does not perform as well as LoRA when using the same BLM. The limitation arises be-

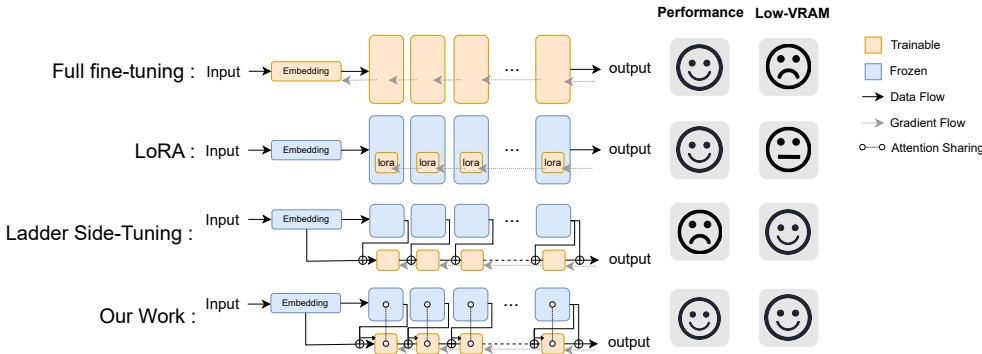

Figure 1: Overview of different methods: Our approach employs an attention-sharing mechanism, achieving competitive performance compared to LST, while maintaining both efficiency and flexibility.

cause the parameters and attention weights in the BLM remain fixed, making it difficult to construct internal representations or adapt the model to task-specific nuances. As a result, for side-tuning methods to outperform other techniques like LoRA, they typically need to operate under similar VRAM constraints but with a larger BLM to compensate for the lack of adaptability in the frozen parameters.

To address the aforementioned issues, we propose Symbiotic tuning (SymTune), which selectively filters significant values from the hidden states of the BLM and integrates an additional contextual processing module to enhance performance. Our work introduces two key operations: Inverse Cross-Attention (ICA) and Attention Sharing (ATS). ICA involves designing a specialized cross-attention mechanism that constructs task-specific representations while preserving the rich contextual information from the backbone model. ATS aims to share additional internal signals from the backbone language model (BLM), enabling the symbiotic module to better learn both the language and the downstream task. Experimental results demonstrate that a language model, when paired with a single symbiotic module containing only a few million parameters, achieves competitive validation scores across a range of natural language understanding benchmarks and multi-label tasks. Moreover, it retains the low VRAM requirements characteristic of side-tuning methods, offering a resource-efficient solution without compromising task-specific performance.

## 2 RELATED WORK

Our method is inspired by contemporary Parameter-Efficient Fine-Tuning (PEFT) and side-tuning techniques, such as LoRA (Hu et al., 2022), which have gained widespread adoption in the research community and demonstrated strong results.

This study focuses on two main aspects: 1) improving task performance, and 2) minimizing the computational costs of training. To this end, we conducted a comprehensive review of PEFT methods, systematically comparing their respective strengths and limitations to inform the development of our approach.

### 2.1 LORA SERIES: HIGH-PERFORMANCE PEFT APPROACHES

Transformer-based models exhibit great performance owing to the robust feature processing capabilities of the attention mechanism and their expansive model structures.

The attention weights $\alpha_{ij}$ are computed as the softmax outputs of the dot product between the $i^{th}$ query $q_i$ and the $j^{th}$ key $k_j$. The attention weights serve as weights for summarizing $v$.

$$\boldsymbol{o}_{\text{lora}} = \boldsymbol{W}\boldsymbol{x} + \Delta\boldsymbol{W}\boldsymbol{x} = \boldsymbol{W}\boldsymbol{x} + \boldsymbol{B}\boldsymbol{A}\boldsymbol{x} \tag{1}$$

LoRA, shown in Eq. 1, introduces additional parameters to learn the adjustments of parameters in the model which are typically $\boldsymbol{W}_q$, $\boldsymbol{W}_v$, and $\boldsymbol{W}_o$ in practical implementations. These modifications influence every critical step in the attention layer processing procedure while requiring a much smaller number of parameters to be adjusted. Consequently, LoRA outperforms full fine-tuning on numerous natural language understanding benchmarks. This superiority stems from the fact that datasets for these benchmarks typically contain limited amounts of data. Under such constraints, it is easier to tune the model when fewer parameters need to be modified.

The success of these PEFT methods and some other inspiring research (Song et al., 2023; Fedus et al., 2022; Chen et al., 2023) offers valuable insights: fine-tuning a language model often involves reconstructing internal representations and adjusting projection matrices of the model, typically under constraints related to data availability. Reducing the number of trainable parameters has proven to be an effective strategy for facilitating optimal convergence, as it simplifies the optimization process and reduces the risk of overfitting when working with limited data.

However, LoRA is unable to reduce the VRAM usage associated with storing activations, which must be kept in VRAM for use during gradient back-propagation to compute partial derivatives. In LoRA, the activation should still be kept to compute the gradient for every $\Delta \boldsymbol{W}$. Consequently, the VRAM required for these activations, which can be substantial in large-scale models, remains unchanged. Therefore, while LoRA is highly effective in reducing the memory required for storing gradients and optimizer states, it does not alleviate the overall VRAM burden significantly.

## 2.2 Side-Tuning: PEFT Approach with Single-Directional Information Flow

Ladder Side-Tuning (LST) (Sung et al., 2022) constructs a new side model based on the hidden states of the BLM. Unlike LoRA, the information flow in LST is unidirectional which flows only from the BLM to the side network. This approach does not alter the internal representations within the BLM, demonstrating that it's still possible to achieve good performance by simply building comprehensive language modeling capability on top of the existing hidden states.

Quantized Side-Tuning (QST) is an alternative method that employs a quantization mechanism to reduce the VRAM consumption associated with the BLM's parameters. This approach achieves a reduction in memory usage without substantially compromising performance when compared to LST while offering even greater VRAM savings.

In Side-Tuning, the hidden states are initially mapped to a lower-dimensional space before being fed into the side layer.

$$\widetilde{\boldsymbol{h}}^{(l)} = f_s^{(l)}(\widetilde{\boldsymbol{h}}^{(l-1)} + \boldsymbol{W}_{\text{down}}^{(l-1)} \boldsymbol{h}^{(l-1)}; \boldsymbol{\theta}_{\text{side}}^{(l)}) \tag{2}$$

Here, $\boldsymbol{W}_{\text{down}}^{(l-1)}$ is the down-projection layer, $\boldsymbol{h}_l$ represents the $l^{th}$ hidden state, and $\widetilde{\boldsymbol{h}}^{(l)}$ refers to the outputs of the $l^{th}$ side layer.

In side-tuning methods, the BLM's weights are frozen, meaning no gradients are computed for the backbone, and consequently, the activations from the BLM do not need to be stored for back-propagation. Instead, the side network, which is much smaller and runs in parallel with the backbone, is the only part of the model where activations need to be retained for gradient calculations. Since the side network is lightweight compared to the full model, the memory required for activations is significantly reduced.

However, side-tuning often involves a trade-off between efficiency and performance. Unlike LoRA, which directly modifies the attention score computation, side-tuning methods rely heavily on the BLM. While initializing the side network with pruned weights from the BLM can enhance training efficiency, its performance on benchmarks like GLUE tends to be inferior to LoRA's when using the same BLM. This is primarily due to its reliance on the BLM, which only provides hidden states without sharing additional internal signals. As a result, LST must establish additional connections to the BLM for reconstructing internal representations and achieve better model performance.

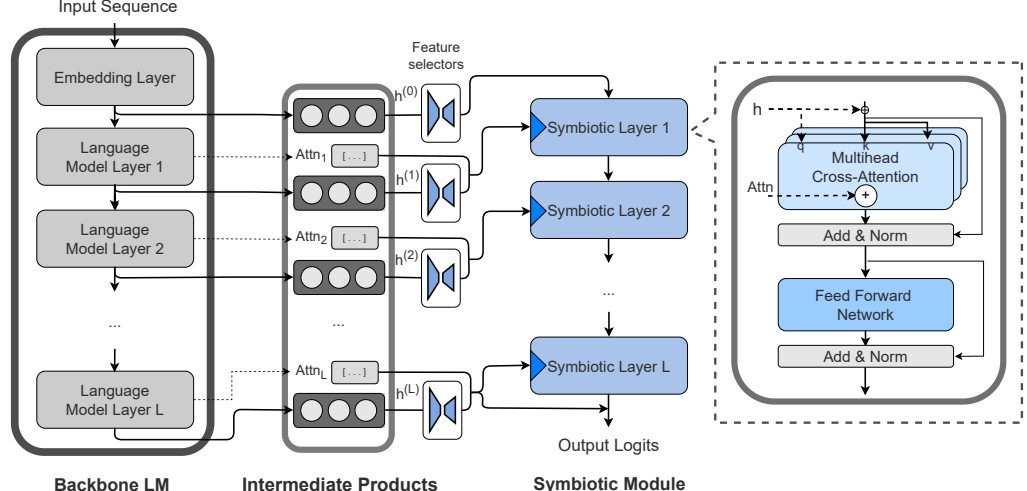

Figure 2: Overall architecture design of SymTune with a pretrained LM. All hidden states and attention weights of the language model are extracted and then used as input of the cross-attention layers in the symbiotic modules, serving as the query part.

## 3 METHODOLOGY

In order to make Symbiotic Tuning adaptable to encoder and decoder of the language models, we respectively introduce symbiotic module construction of encoder and decoder in section 3.1 and section 3.2. Besides, we briefly introduce the forward pass and back-propagation in section 3.3.

### 3.1 ENCODER MODEL CONSTRUCTION

Our model is designed as Figure 2. We define the hidden states in the $L$-layer BLM as $\mathbb{H} = \{\boldsymbol{h}^{(0)}, \boldsymbol{h}^{(1)}, \boldsymbol{h}^{(2)}, ..., \boldsymbol{h}^{(L)}\}$, where $\boldsymbol{h}^{(0)}$ represents the output of the embedding layer, and the attention weights as $\mathbb{A} = \{\boldsymbol{A}^{(1)}, \boldsymbol{A}^{(2)}, ..., \boldsymbol{A}^{(L)}\}$. The outputs of the $L$-layer symbiotic module are defined as $\widetilde{\mathbb{H}} = \{\widetilde{\boldsymbol{h}}^{(1)}, \widetilde{\boldsymbol{h}}^{(2)}, ..., \widetilde{\boldsymbol{h}}^{(L)}\}$. The data dimensions of the hidden states and the symbiotic module are represented by $d_h$ and $d_{\widetilde{h}}$, respectively. Furthermore, there is a feature selector which consists of a down projection and an up projection, denoted as $\boldsymbol{W}_A \boldsymbol{W}_B$, where $\boldsymbol{W}_A \in \mathbb{R}^{d_h \times r}$, $\boldsymbol{W}_B \in \mathbb{R}^{r \times d_{\widetilde{h}}}$. We simplify $\boldsymbol{W}_A \boldsymbol{W}_B$ to $\boldsymbol{W}_{AB}$.

As shown in Figure 2, we obtain intermediate products from BLM which consist of all of the hidden states and attention weights. Unlike LST which is shown in Eq. 2, our symbiotic tuning is like:

$$\widetilde{\boldsymbol{h}}^{(l)} = f_{S.T.}^{(l)}(\widetilde{\boldsymbol{h}}^{(l-1)}, \boldsymbol{A}^{(l)}, \boldsymbol{W}_{AB}^{(l)} \boldsymbol{h}^{(l)}; \boldsymbol{\theta}_{S.T.}^{(l)}) \tag{3}$$

$$\widetilde{\boldsymbol{h}}^{(0)} = \boldsymbol{W}_{AB}^{(0)} \boldsymbol{h}^{(0)}$$

The symbiotic modules, which consist of a feature selector and a tiny transformer, rely on the intermediate products and operate in a low-dimensional space. Therefore, the purpose of the feature selector is to dynamically identify and select the crucial parameters. Meanwhile, the symbiotic module is responsible for performing natural language feature processing, leveraging the selected parameters to adapt to specific downstream tasks efficiently. This dynamic selection process ensures that only the most relevant information is utilized for each task, contributing to the overall effectiveness of the approach. Each layer of the symbiotic module consists of two parts: multi-head cross-attention, and a feed-forward network.

### 3.1.1 INVERSE CROSS ATTENTION (ICA)

The hidden state $\boldsymbol{h}_l$ is first linearly projected into a lower-dimensional symbiotic space by a feature selector, which functions as a form of continuous pruning that dynamically selects the most crucial

values from the hidden states. To construct task-specific internal representations and information flow while referencing the hidden states from the BLM, we utilize the hidden states of the language model as queries, rather than as keys or values. This approach is the inverse of the typical transformer decoder mechanism which takes the outputs of previous decoder layers as query. With ICA, the model is allowed to dynamically connect the hidden states of BLM, which facilitates the construction of internal representations specific to the downstream task. Besides, inspired by Zhang et al. (2024), we apply a weighting mechanism to the keys and values in the cross-attention layer. This mechanism dynamically combines the backbone hidden states $\boldsymbol{h}^{(l)}$ with the symbiotic layer hidden states $\widetilde{\boldsymbol{h}}^{(l)}$. The queries $Q$, keys $K$ and values $V$ are shown in Eq. 4.

$$\boldsymbol{Q}^{(l)} = \boldsymbol{W}_q^{(l)}(\boldsymbol{W}_{AB}^{(l)}\boldsymbol{h}^{(l)}) \quad \boldsymbol{K}^{(l)} = \boldsymbol{W}_k^{(l)}\widetilde{\boldsymbol{h}}_{KV}^{(l)} \quad \boldsymbol{V}^{(l)} = \boldsymbol{W}_v^{(l)}\widetilde{\boldsymbol{h}}_{KV}^{(l)} \tag{4}$$

$$\widetilde{\boldsymbol{h}}_{KV}^{(l)} = c^{(l)}\boldsymbol{W}_{AB}^{(l)}\boldsymbol{h}^{(l)} + (1 - c^{(l)})\widetilde{\boldsymbol{h}}^{(l-1)}$$

Here, $c^{(l)} \in (0, 1)$ is a trainable weight parameter for layer $l$. It is initially set to 0.5 and subsequently adjusted during the training process.

### 3.1.2 ATTENTION SHARING (ATS)

The attention weights are integrated with those from the BLM to stabilize training and improve performance. This allows for dynamic and fine-grained adjustments to information among internal representations. By selectively focusing on key features within the hidden states, the model establishes more adaptable, context-aware interactions that enable it to better utilize internal representations from the BLM. This combined attention mechanism facilitates a more robust and flexible learning process, effectively leveraging the pre-trained knowledge of the backbone while refining task-specific representations. Our inverse cross-attention $f_{ICA}$ is formalized as:

$$f_{\text{ICA}}^{(l)}(\boldsymbol{Q}^{(l)}, \boldsymbol{K}^{(l)}, \boldsymbol{V}^{(l)}) = (\text{Softmax}(\frac{\boldsymbol{Q}^{(l)}\boldsymbol{K}^{(l)\top}}{\sqrt{d_{\widetilde{h}}}}) + \boldsymbol{A}^{(l)} - \boldsymbol{B})\boldsymbol{V}^{(l)} \tag{5}$$

Where $\boldsymbol{Q}, \boldsymbol{K}, \boldsymbol{V}$ represent the query, key, and value matrices, and $\boldsymbol{A}$ indicates attention weights from BLM. To maintain the normalization of attention weights (i.e., ensuring that the sum of attention values equals 1), we introduce a balancing matrix $\boldsymbol{B}$ in Eq. 5. This matrix consists of elements defined as $1/s$, where $s$ represents the length of the input sequence.

### 3.1.3 WHY DO WE NEED ATS?

As previously stated, side-tuning methods such as LST and QST adapt the outputs of a BLM without sharing the internal signals of the Transformer modules. Although side-tuning reduces memory consumption during training by leveraging side networks in the back-propagation process, the absence of internal connections between the side networks and the BLM can lead to suboptimal task performance. In contrast, our approach, Symbiotic Tuning, integrates attention weights from the BLM and introduces a novel attention mechanism, offering a unified attention flow between the BLM and side networks for enhancing tuning performance.

## 3.2 DECODER MODEL CONSTRUCTION

For decoder-only models, the overall architecture of the symbiotic modules remains consistent with that used in encoder-only models. However, a critical distinction arises in the use of attention-masking mechanisms specific to decoder architectures. In these models, each token can only compute attention weights based on its preceding tokens, which is essential for preserving the auto-regressive nature of the decoder. This masking ensures that future tokens do not influence the current token during training or inference. As a result, the attention-sharing mechanism between the backbone and the symbiotic modules must account for this feature, otherwise, it could introduce unwanted information flow or noise, which would disrupt the model's ability to learn meaningful internal representations within the masked attention framework. The cross attention is shown in Eq. 6.

$$f_{\text{ICA}-\text{decoder}}^{(l)}(\boldsymbol{Q}^{(l)}, \boldsymbol{K}^{(l)}, \boldsymbol{V}^{(l)}) = (\text{Softmax}(\frac{\boldsymbol{Q}^{(l)}\boldsymbol{K}^{(l)\top}}{\sqrt{d_{\tilde{h}}}}) + \boldsymbol{A}^{(l)} - \boldsymbol{B})\boldsymbol{V}^{(l)} \qquad (6)$$

$$\boldsymbol{B} = [b_{ij}] = \begin{pmatrix} b_{11} & b_{12} & \cdots & b_{1s} \\ b_{21} & b_{22} & \cdots & b_{2s} \\ \vdots & \vdots & \vdots & \vdots \\ b_{s1} & b_{s2} & \cdots & b_{ss} \end{pmatrix} \quad \text{, where} \quad b_{ij} = \begin{cases} 0, & \text{if } i < j \\ \frac{1}{i}, & \text{otherwise} \end{cases}$$

As described in 3.1.2, the balancing matrix $\boldsymbol{B}$ ensures that the masked elements in the attention weights remain 0 while simultaneously normalizing each row so that the sum of the attention weights equals 1.

## 3.3 Forward and Backward Pass: Side-tuning and Back Propagation

Unlike LST (Sung et al., 2022) and QST (Zhang et al., 2024), our symbiotic module is not a smaller version of the BLM. Instead, the first self-attention layer is replaced with an inverse cross-attention. During the forward pass, the BLM operates in evaluation mode, meaning no dropout, batch normalization, or activation storage is applied, and no gradients are computed for the BLM.

As shown in Figure 1, during the backward pass, gradients are propagated solely through the symbiotic module, while the BLM remains unaffected by gradient updates. This inference-only mode for the backbone significantly reduces the VRAM usage of our method, as it eliminates the need to store activations and reduces memory requirements during model training.

## 4 Experiments

For each aforementioned aspect in which we aim to make improvements, we design experiments to demonstrate its robustness individually. In encoder model experiments and decoder model experiments, we compare our model with high-performance PEFT approaches, side-tuning approaches, and full fine-tuned language models. In multi-label experiments, we compare our methods with other fine-tuning approaches including LoRA and LST. Additionally, we measure the transmission latency in distribution experiments. We conduct experiments using five random seeds and report the average scores on the validation set. We configured the hidden size to 96 and set the rank to 8 as the default setting to evaluate performance across various benchmarks. The hyper-parameter settings of encoder-only model experiments, decoder-model experiments and multi-label experiments are presented in Appendix B.

### 4.1 Encoder Model Experiments

Table 1: Performance experiments on natural language understanding tasks. We compare our methods with current LoRA-based PEFT methods on benchmark datasets. The best results on each benchmark are shown in **bold**. Furthermore, we compute the p-values for all baselines with our approach to assess whether our approach significantly outperforms them. A p-value less than 0.05, highlighted in **bold**, indicates a statistically significant difference.

| / | # Params | CoLA | SST2 | MRPC | STSB | QQP | RTE | QNLI | MNLI | Avg. | P-value |
|---|---|---|---|---|---|---|---|---|---|---|---|
| metrics | / | Mcc | Acc | Acc | Corr | Acc/F1 | Acc | Acc | m/mm | / | / |
| BLM: DeBERTaV3-base, batch size = 16 | | | | | | | | | | | |
| FFT | 183m | $69.2_{\pm0.6}$ | $95.6_{\pm0.8}$ | $89.5_{\pm0.6}$ | $91.6_{\pm0.4}$ | $91.7_{\pm0.3}/89.7_{\pm0.4}$ | $83.7_{\pm1.1}$ | $93.7_{\pm0.3}$ | $89.4_{\pm0.2}/89.9_{\pm0.1}$ | 88.2 | 0.3025 |
| Bitfit | 0.1m | $67.2_{\pm0.7}$ | $93.6_{\pm0.5}$ | $87.7_{\pm1.0}$ | $90.3_{\pm0.2}$ | $88.7_{\pm0.6}/85.0_{\pm0.4}$ | $78.4_{\pm1.3}$ | $92.4_{\pm0.2}$ | $87.4_{\pm0.1}/87.7_{\pm0.1}$ | 85.9 | **0.0149** |
| LoRA (r=8) | 0.8m | $69.4_{\pm0.8}$ | $95.8_{\pm0.6}$ | $90.7_{\pm0.7}$ | $91.1_{\pm0.3}$ | $\mathbf{90.3}_{\pm0.4}/87.4_{\pm0.5}$ | $87.7_{\pm1.4}$ | $93.1_{\pm0.4}$ | $\mathbf{88.9}_{\pm0.3}/\mathbf{89.3}_{\pm0.2}$ | 88.5 | **0.0184** |
| LST | 1.8m | $69.0_{\pm0.4}$ | $95.7_{\pm0.4}$ | $91.0_{\pm0.5}$ | $91.2_{\pm0.4}$ | $89.8_{\pm0.5}/88.9_{\pm0.3}$ | $88.6_{\pm0.9}$ | $92.9_{\pm0.5}$ | $88.4_{\pm0.3}/88.5_{\pm0.3}$ | 88.3 | **0.0041** |
| S.T. (ours) | 0.9m | $\mathbf{70.1}_{\pm0.6}$ | $\mathbf{95.8}_{\pm0.3}$ | $\mathbf{91.2}_{\pm0.5}$ | $\mathbf{91.5}_{\pm0.3}$ | $90.0_{\pm0.5}/\mathbf{89.5}_{\pm0.3}$ | $\mathbf{89.9}_{\pm1.3}$ | $\mathbf{93.6}_{\pm0.3}$ | $\mathbf{88.9}_{\pm0.1}/89.1_{\pm0.2}$ | **88.9** | / |
| BLM: DeBERTaV3-large, batch size = 16 | | | | | | | | | | | |
| FFT | 434m | $74.2_{\pm0.5}$ | $95.9_{\pm0.4}$ | $92.1_{\pm0.3}$ | $92.7_{\pm0.4}$ | $91.2_{\pm0.1}/91.2_{\pm0.2}$ | $90.4_{\pm0.9}$ | $95.1_{\pm0.2}$ | $90.9_{\pm0.4}/91.0_{\pm0.3}$ | 90.4 | 0.3796 |
| Bitfit | 0.1m | $70.9_{\pm1.0}$ | $\mathbf{96.2}_{\pm0.2}$ | $90.4_{\pm0.8}$ | $91.3_{\pm0.5}$ | $89.2_{\pm0.4}/86.0_{\pm0.5}$ | $87.7_{\pm1.6}$ | $94.4_{\pm0.4}$ | $\mathbf{91.2}_{\pm0.6}/\mathbf{91.0}_{\pm0.3}$ | 89.1 | **0.0052** |
| LoRA (r=16) | 2.6m | $72.9_{\pm0.5}$ | $96.1_{\pm0.3}$ | $91.1_{\pm0.3}$ | $92.2_{\pm0.3}$ | $\mathbf{92.2}_{\pm0.5}/90.5_{\pm0.6}$ | $90.9_{\pm1.1}$ | $95.0_{\pm0.6}$ | $90.3_{\pm0.2}/90.7_{\pm0.2}$ | 90.2 | 0.3198 |
| LST | 4.3m | $71.1_{\pm0.9}$ | $96.0_{\pm0.3}$ | $91.7_{\pm0.5}$ | $91.8_{\pm0.3}$ | $90.6_{\pm0.3}/90.2_{\pm0.4}$ | $91.0_{\pm1.2}$ | $94.3_{\pm0.3}$ | $89.8_{\pm0.1}/90.4_{\pm0.3}$ | 89.7 | **0.0144** |
| S.T. (ours) | 2.2m | $\mathbf{73.2}_{\pm0.7}$ | $96.1_{\pm0.6}$ | $\mathbf{92.3}_{\pm0.6}$ | $\mathbf{92.6}_{\pm0.3}$ | $90.9_{\pm0.2}/\mathbf{90.6}_{\pm0.3}$ | $\mathbf{91.7}_{\pm1.1}$ | $\mathbf{94.7}_{\pm0.5}$ | $90.2_{\pm0.3}/90.6_{\pm0.2}$ | **90.3** | / |

Table 2: VRAM requirements (GB) on training each natural language understanding tasks. We standardized the batch size to 16 across all baselines and benchmarks. The VRAM requirements vary depending on the input sequence lengths.

| / | # Params | CoLA | SST2 | MRPC | STSB | QQP | RTE | QNLI | MNLI | Avg. |
|---|---|---|---|---|---|---|---|---|---|---|
| Avg. length | / | 7.7 | 9.4 | 45.9 | 21.9 | 24.1 | 54.4 | 38.5 | 31.8 | / |
| BLM: DeBERTaV3-base, batch size = 16 | | | | | | | | | | |
| FFT | 183m | 5.7 | 6.1 | 6.6 | 8.80 | 12.9 | 13.8 | 14.1 | 6.7 | 9.3 |
| Bitfit | 0.1m | 3.4 | 4.1 | 8.1 | 10.3 | 12.0 | 14.6 | 14.3 | 7.0 | 9.2 |
| LoRA (r=8) | 0.8m | 3.2 | 3.4 | 4.7 | 4.1 | 7.2 | 7.3 | 14.8 | 8.9 | 6.7 |
| LST | 1.8m | 2.2 | 2.3 | 2.6 | 2.8 | 5.7 | 6.4 | 10.6 | 3.7 | 4.5 |
| S.T. (ours) | 0.9m | **2.2** | **2.3** | **2.5** | **2.8** | **5.6** | **6.0** | **11.1** | **3.1** | **4.5** |
| BLM: DeBERTaV3-large, batch size = 16 | | | | | | | | | | |
| FFT | 434m | 13.7 | 13.5 | 20.8 | 22.7 | 34.2 | 41.1 | 36.2 | 15.6 | 24.7 |
| Bitfit | 0.3m | 12.4 | 13.9 | 15.2 | 22.5 | 20.3 | 34.7 | 35.8 | 14.9 | 21.2 |
| LoRA (r=16) | 2.6m | 5.6 | 6.0 | 7.1 | 7.8 | 15.5 | 15.3 | 33.1 | 19.8 | 13.8 |
| LST | 4.3m | 4.2 | 4.4 | 5.2 | 6.0 | 13.1 | 14.7 | 19.3 | 4.5 | 8.9 |
| S.T. (ours) | 2.2m | **4.2** | **4.3** | **5.2** | **6.0** | **13.1** | **14.7** | **19.2** | **4.4** | **8.9** |

We compared our method, which incorporates a single symbiotic module, with LoRA (Hu et al., 2022), Bitfit (Ben Zaken et al., 2022), LST (Sung et al., 2022), and the full fine-tuning method (FFT) on the GLUE (Wang et al., 2018) benchmark. Details including the task types and data sizes can be found in Appendix A.

We employ both the base and large versions of DeBERTa-V3 (He et al., 2023) as the BLMs to evaluate natural language understanding performance across benchmarks that cover different aspects of the task. The results, as presented in Table 1 and Table 2, demonstrate that our method performs well across various datasets while maintaining minimal VRAM requirements for almost all benchmarks. Notably, our approach excels in smaller datasets like CoLA, STSB, and RTE, where the architecture's efficient handling of internal representations plays a crucial role. On larger datasets, such as QQP, MNLI, and QNLI, the greater data volume provides enough information to reconstruct internal representations, enabling full fine-tuning to yield optimal performance.

The statistical significance test results are also shown in Table 1. The p-values for LST are well below the threshold of 0.05, particularly for the larger 2.7B model, indicating that our approach significantly outperforms LST. For LoRA, our method achieves slightly better performance while requiring substantially less VRAM, highlighting its ability to enhance side-tuning methods, deliver superior results, and effectively compete with LoRA on the same BLM.

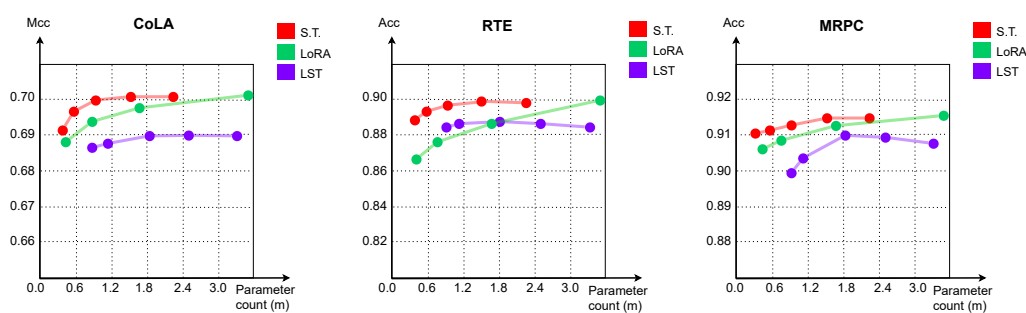

Figure 3: Experiments on different trainable parameter counts. We employed various configurations with different parameter counts to compare performance outcomes on DeBERTaV3-base. The parameter count and VRAM requirement were controlled by adjusting the hidden dimension and projection ranks, where the hidden dimension $d_{\widetilde{h}} \in \{60, 72, 96, 120, 144\}$ the projection ranks $r \in \{4, 8, 10, 12, 16\}$. This provides a clearer understanding of how changes in parameterization influence overall performance.

We further conducted experiments using symbiotic modules with varying parameter counts on CoLA, RTE, and MRPC to assess their performance. As illustrated in Figure 3 and Figure 4, our symbiotic module consistently performs better across both low and high parameter count settings.

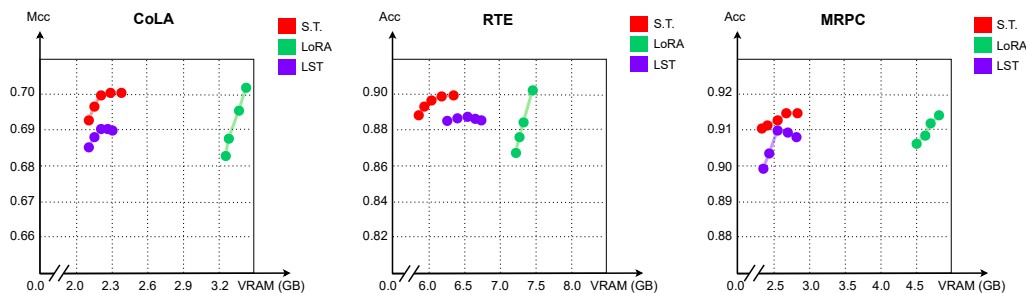

Figure 4: Experiments on different VRAM requirements. We compare the performance variations across different VRAM settings to provide a clearer understanding of how our method outperforms LST and LoRA. Compared to LoRA, our method significantly reduces VRAM requirements, and compared to LST, our evaluation scores are consistently higher.

Its stability across different parameter scales highlights its adaptability. These results underscore the robustness and efficiency of our architecture, effectively balancing performance with computational cost.

## 4.2 DECODER MODEL EXPERIMENTS

The experiments on the decoder-only model are summarized in Table 3, where we use QST and LST as baseline models. These baselines are fine-tuned on the OPT-1.3b and OPT-2.7b models (Zhang et al., 2022) using the GLUE benchmark (Wang et al., 2018). Statistical significance tests, also presented in Table 3, confirm that the observed performance improvements are not due to random bias. The results demonstrate that Symbiotic Tuning outperforms LST on nearly all datasets with statistically significant differences. Furthermore, the performance of Symbiotic Tuning on GLUE benchmarks with OPT-1.3b is very close to QST's performance on OPT-2.7b, while requiring 4 GB less VRAM. The inference times for BLM, LST, and our method are provided in Appendix C.

Table 3: Performance experiments on decoder-only model. We compare our methods on OPT-series model with QST and LST, and their experiment results are from Zhang et al. (2024). We report the trainable parameter count for each baseline, along with the VRAM consumption during training with a batch size of 16 and a maximum sequence length of 512. The best results on each benchmark are shown in **bold**. Besides, p-values for all the baselines with our method are computed, with values less than 0.05 highlighted in **bold**.

| /
metrics | # Params
/ | VRAM
GB | CoLA
Mcc | SST2
Acc | MRPC
Acc | STSB
Corr | QQP
Acc | RTE
Acc | QNLI
Acc | MNLI
Acc | Avg.
/ | P-value
/ |
|---|---|---|---|---|---|---|---|---|---|---|---|---|
| | | | | | BLM: OPT-1.3b, batch size = 16 | | | | | | | |
| LoRA | 31.0m | 32.9 | $62.5_{\pm 1.7}$ | $93.7_{\pm 0.7}$ | $\mathbf{83.4}_{\pm 0.9}$ | $\mathbf{89.3}_{\pm 0.2}$ | $86.9_{\pm 0.3}$ | $82.7_{\pm 1.9}$ | $81.4_{\pm 9.3}$ | $81.2_{\pm 0.1}$ | 82.6 | 0.7598 |
| QST | 5.9m | 17.7 | $59.7_{\pm 2.9}$ | $94.4_{\pm 0.3}$ | $81.7_{\pm 1.1}$ | $88.4_{\pm 1.1}$ | $84.3_{\pm 0.7}$ | $79.5_{\pm 2.5}$ | $85.7_{\pm 0.5}$ | $77.1_{\pm 0.6}$ | 81.3 | **0.0030** |
| LST | 31.4m | 20.9 | $59.5_{\pm 3.1}$ | $95.2_{\pm 0.8}$ | $83.1_{\pm 1.3}$ | $88.6_{\pm 0.4}$ | $86.4_{\pm 0.6}$ | $\mathbf{82.0}_{\pm 2.2}$ | $86.1_{\pm 0.3}$ | $77.8_{\pm 0.5}$ | 82.2 | 0.1747 |
| S.T. | 5.9m | 20.7 | $60.3_{\pm 1.7}$ | $\mathbf{95.3}_{\pm 0.7}$ | $83.3_{\pm 1.0}$ | $88.6_{\pm 0.7}$ | $\mathbf{87.2}_{\pm 0.6}$ | $79.8_{\pm 2.4}$ | $\mathbf{87.5}_{\pm 0.6}$ | $\mathbf{80.8}_{\pm 0.4}$ | **82.8** | / |
| | | | | | BLM: OPT-2.7b, batch size = 16 | | | | | | | |
| LoRA | 51.3m | 50.4 | $\mathbf{64.5}_{\pm 2.4}$ | $95.3_{\pm 0.6}$ | $\mathbf{84.6}_{\pm 0.8}$ | $\mathbf{90.9}_{\pm 0.1}$ | $\mathbf{90.7}_{\pm 0.1}$ | $\mathbf{82.7}_{\pm 1.9}$ | $83.0_{\pm 7.4}$ | $82.6_{\pm 0.2}$ | 84.5 | 0.8381 |
| QST | 11.8m | 24.4 | $62.0_{\pm 3.4}$ | $94.3_{\pm 0.3}$ | $83.7_{\pm 1.2}$ | $88.9_{\pm 1.4}$ | $86.5_{\pm 0.9}$ | $80.1_{\pm 2.1}$ | $86.6_{\pm 0.9}$ | $80.4_{\pm 0.6}$ | 83.0 | **0.0152** |
| LST | 64.5m | 30.7 | $60.7_{\pm 3.5}$ | $95.3_{\pm 0.4}$ | $83.9_{\pm 1.5}$ | $89.1_{\pm 0.9}$ | $88.8_{\pm 1.0}$ | $82.5_{\pm 2.9}$ | $87.3_{\pm 0.2}$ | $80.4_{\pm 0.7}$ | 83.5 | **0.0045** |
| S.T. | 11.8m | 28.9 | $61.0_{\pm 1.9}$ | $\mathbf{96.2}_{\pm 0.5}$ | $84.1_{\pm 1.1}$ | $89.3_{\pm 0.7}$ | $90.0_{\pm 0.9}$ | $80.5_{\pm 2.6}$ | $\mathbf{89.2}_{\pm 0.8}$ | $\mathbf{85.6}_{\pm 0.8}$ | **84.5** | / |

## 4.3 MULTI-LABEL EXPERIMENTS

To evaluate the performance of our approach in a more challenging scenario, we apply it to multi-label classification tasks for further assessment. We conduct a comprehensive comparative analysis with the full fine-tuning approach and LST. Traditional approaches typically utilize a shared BLM along with multiple task-specific projection modules, usually in the form of classifiers. We evaluate our method using SemEval2014-Task1 (Marelli et al., 2014) and SemEval2018-Task1 (Mohammad et al., 2018) as benchmarks. The experimental results are presented in Table 4 and Table 5.

SemEval2014-Task1 is utilized for both textual entailment recognition and relatedness score prediction, while SemEval2018-Task1 serves as a sentiment analysis dataset with 11 classes.

As shown in Table 4 and Table 5, SymTune consistently performs well across most subtasks. This is particularly evident when the subtasks are less similar, such as in SemEval2014-Task1. While relatedness and entailment share some commonalities, they differ in nature—relatedness is a regression task, and entailment is classification. In FFT, both tasks rely on the same hidden states, potentially limiting performance. SymTune, however, constructs independent representations for each subtask, significantly boosting performance, particularly in tasks with distinct characteristics.

Table 4: Multi-label experiments on SemEval2014 Task1. We adopt multitask fine-tuning to our SymTune with multiple symbiotic module, The best results are shown in **bold**.

| task metrics | **#Params /** | relatedness Corr | entailment Acc |
|---|---|---|---|
| FFT | 183m | $90.36_{\pm 0.20}$ | $89.46_{\pm 0.33}$ |
| LST | 1.8m | $91.84_{\pm 0.30}$ | $90.28_{\pm 0.15}$ |
| S.T. | 0.9m | $\mathbf{92.04}_{\pm 0.40}$ | $\mathbf{90.76}_{\pm 0.27}$ |

SemEval2018-Task1 is a multi-label classification problem requiring the model to identify multiple emotions in the text. We utilize this dataset to test multitask learning performance. The lowest score is consistently "surprise" due to the insufficient number of positive samples for effective learning. Despite this, the experiments demonstrate that SymTune performs well on this sentiment analysis task.

The advantages of our approach primarily include: 1) Despite employing multiple symbiotic modules for these subtasks, the total number of trainable parameters remains significantly lower than in full fine-tuned multitask learning (FFT). This is because the base language model remains frozen in SymTune, whereas it needs to be tuned in the full fine-tuning method. 2) When a new task is introduced to the system, the training cost of SymTune is significantly lower compared to FFT (full fine-tuning). Because in FFT, we must conduct multitask learning for all subtasks again to adapt the base language model to all tasks simultaneously. In contrast, with SymTune, we only need to train an additional symbiotic module. This is because the computation process of a symbiotic module has no impact on the BLM or other symbiotic modules. 3) The integration of attention score shortcuts with

Table 5: Multi-label experiments on SemEval2018 Task1. We report macro F1 scores across the 11 separate sentiment analysis tasks. The best results are shown in **bold**.

| emotions / | **FFT** 183m | **LST** 1.8m | **S.T.** 0.9m |
|---|---|---|---|
| anger | $83.69_{\pm 0.40}$ | $84.37_{\pm 0.21}$ | $\mathbf{85.48}_{\pm 0.19}$ |
| anticipation | $61.67_{\pm 0.48}$ | $64.37_{\pm 0.60}$ | $\mathbf{65.50}_{\pm 0.46}$ |
| disgust | $80.21_{\pm 0.31}$ | $82.44_{\pm 0.30}$ | $\mathbf{83.52}_{\pm 0.32}$ |
| fear | $82.42_{\pm 0.22}$ | $84.53_{\pm 0.28}$ | $\mathbf{84.87}_{\pm 0.60}$ |
| joy | $87.00_{\pm 0.79}$ | $87.26_{\pm 0.37}$ | $\mathbf{88.44}_{\pm 0.41}$ |
| love | $77.29_{\pm 0.43}$ | $78.60_{\pm 0.55}$ | $\mathbf{79.38}_{\pm 0.33}$ |
| optimism | $81.28_{\pm 0.48}$ | $81.94_{\pm 0.50}$ | $\mathbf{82.33}_{\pm 0.53}$ |
| pessimism | $61.64_{\pm 0.59}$ | $65.13_{\pm 0.77}$ | $\mathbf{67.75}_{\pm 0.70}$ |
| sadness | $76.81_{\pm 0.23}$ | $78.08_{\pm 0.27}$ | $\mathbf{78.42}_{\pm 0.25}$ |
| surprise | $68.86_{\pm 0.61}$ | $67.91_{\pm 0.52}$ | $\mathbf{69.20}_{\pm 0.90}$ |
| trust | $55.49_{\pm 0.63}$ | $58.15_{\pm 1.20}$ | $\mathbf{58.64}_{\pm 1.18}$ |
| Avg. | $74.21_{\pm 0.32}$ | $75.91_{\pm 0.24}$ | $\mathbf{76.69}_{\pm 0.27}$ |

cross-attention layers in the Symbiotic modules enhances the performance across multiple subtasks, while simultaneously reducing the overall parameter count. This design not only improves task-specific outcomes but also positions SymTune as a more efficient alternative to LST, eliminating the traditional trade-off between efficiency and performance. Additionally, these symbiotic modules can be trained with different hyper parameters such as learning rates, warmup strategies, learning rate schedulers, or even different optimizers. This flexibility enables us to optimize training for each subtask, potentially leading to better overall performance.

## 4.4 ABLATION STUDY

We introduced three novel operations in our model: Inverse Cross Attention (ICA) and Attention Sharing (ATS). Abandoning the ICA reverts the model to the LST approach, where the hidden states of the BLM are directly added to the output of the symbiotic module's final layer.

The ablation study results, which are shown in Table 6, indicate that both Cross Attention and Attention Sharing provide substantial improvements in the language model's performance across a wide range of benchmarks. The experimental results indicate that just using ICA does not significantly impact the outcomes. However, combined with ATS, it leads to noticeable improvements in the GLUE benchmark. Additionally, the findings demonstrate that sharing attention scores is a reasonable and compatible approach for enhancing natural language understanding capabilities.

Table 6: Ablation study. We respectively remove each of our operations to evaluate their effect on performances. The best results on each benchmark are shown in **bold**. Besides, p-values are computed, with values less than 0.05 highlighted in **bold**.
.

| / | CoLA | SST2 | MRPC | STSB | QQP | RTE | QNLI | MNLI | Avg. | P-value |
|---|---|---|---|---|---|---|---|---|---|---|
| metrics | Mcc | Acc | Acc | Corr | Acc/F1 | Acc | Acc | m/mm | / | / |
| S.T. (ours) | $70.1_{\pm0.6}$ | $95.8_{\pm0.3}$ | $91.2_{\pm0.5}$ | $91.5_{\pm0.3}$ | $90.0_{\pm0.5}/89.5_{\pm0.3}$ | $89.9_{\pm1.3}$ | $93.6_{\pm0.3}$ | $88.9_{\pm0.3}/89.1_{\pm0.1}$ | **88.9** | / |
| - ICA | $69.9_{\pm0.7}$ | $95.8_{\pm0.3}$ | $91.1_{\pm0.4}$ | $91.1_{\pm0.4}$ | $89.5_{\pm0.5}/89.0_{\pm0.2}$ | $88.9_{\pm1.3}$ | $93.1_{\pm0.4}$ | $88.8_{\pm0.2}/88.9_{\pm0.2}$ | 88.6 | **0.118** |
| - ATS | $69.2_{\pm0.6}$ | $95.4_{\pm0.4}$ | $90.9_{\pm0.8}$ | $90.8_{\pm0.6}$ | $89.4_{\pm0.3}/88.7_{\pm0.4}$ | $88.4_{\pm1.1}$ | $92.4_{\pm0.3}$ | $88.3_{\pm0.1}/88.6_{\pm0.2}$ | 88.1 | **0.0001** |
| - ICA, ATS | $69.4_{\pm0.5}$ | $95.6_{\pm0.5}$ | $90.6_{\pm0.9}$ | $91.1_{\pm0.3}$ | $89.5_{\pm0.4}/88.9_{\pm0.4}$ | $88.7_{\pm0.7}$ | $92.3_{\pm0.3}$ | $88.2_{\pm0.2}/88.4_{\pm0.3}$ | 88.2 | **0.0003** |

## 5 CONCLUSION

In this work, we introduce Symbiotic Tuning (SymTune), a novel approach that integrates additional symbiotic modules into a language model, providing a robust solution for multitask learning while significantly reducing the overall parameter count. Through extensive experimentation, SymTune has demonstrated a notable reduction in computational costs and GPU memory consumption compared to traditional full fine-tuning and existing PEFT methods. The architecture of SymTune, along with its plug-and-play adaptability, enhances its efficiency and flexibility across various natural language understanding tasks. These results underscore SymTune's potential as a scalable, parameter-efficient framework for multitask learning, offering significant promise for future advancements in natural language understanding.

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

## A    BENCHMARKS STATISTICS

Table 7: Benchmarks statistics. The following is a summary of each dataset used in our experiments, detailing their purposes and associated tasks.

| dataset | discription | task | # samples |
|---|---|---|---|
| CoLA | Linguistic Acceptability | Classification | 8551 / 1043 |
| SST2 | Sentiment Analysis | Classification | 67350 / 873 |
| MRPC | Sentence Equivalence | Classification | 5801 / 408 |
| STSB | Sentence Similarity | Regression | 5712 / 1471 |
| QQP | Paraphrase Recognition | Classification | 363847 / 40431 |
| RTE | Textual Entailment | Classification | 2491 / 278 |
| QNLI | Natural Language Inference | Classification | 103141 / 5268 |
| MNLI | Textual Entailment | Classification | 392702 / 9815 / 9832 |
| SemEval2014Task1 | Sentence Similarity / Textual Entailment | Regression/Classification | 4500 / 500 |
| SemEval2018Task1 | Sentiment Analysis | Multiclass Classification | 6838 / 884 |

Table 7 provides a comprehensive overview of all the datasets utilized in our experiments. The benchmarks for Natural Language Understanding are derived from the General Language Understanding Evaluation (GLUE) benchmark (Wang et al., 2018; Socher et al., 2013; Rajpurkar et al., 2016; Wang et al., 2017; Dagan et al., 2005; Warstadt et al., 2019; El-Said et al., 2015). The SemEval2018-Task1 dataset is designed for multiclass classification and includes multiple classes of emotions. In this task, the model's objective is to accurately identify and select multiple emotions that correspond to the sentiment expressed in the given text. To effectively compare our model's performance with established baselines, We approach it as a multitask learning problem to facilitate comparison with baselines.

## B    HYPER PARAMETERS

We provide the training hyperparameters for encoder-only, decoder-only, and multi-label experiments in Table 8, Table 9, and Table 10, respectively. For encoder-only model experiments, we test 30 different learning rates ranging from 4e-4 to 1.5e-4, selecting the best-performing learning rate to repeat the experiments five times. For decoder-only models, we adopt the learning rate settings from QST Zhang et al. (2024) and conduct experiments under the same conditions as QST and its baselines. For multi-label experiments, we select the learning rate that maximizes the total score across all baselines.

Table 8: Hyper parameters of encoder-only model experiments

| / | CoLA | SST2 | MRPC | STSB | QQP | RTE | QNLI | MNLI |
|---|---|---|---|---|---|---|---|---|
| BLM: DeBERTaV3-base | | | | | | | | |
| learning rate (lr) | 7e-4 | 5e-4 | 1e-3 | 1.3e-3 | 1e-4 | 1.4e-3 | 1.8e-4 | 6e-4 |
| batch size | 16 | 16 | 16 | 16 | 16 | 16 | 16 | 16 |
| lr scheduler | cosine | cosine | cosine | cosine | cosine | cosine | cosine | cosine |
| epochs | 30 | 24 | 50 | 40 | 8 | 50 | 10 | 8 |
| max length | 64 | 128 | 320 | 256 | 320 | 320 | 512 | 256 |
| weight decay | 0.1 | 0.1 | 0.1 | 0.1 | 0.1 | 0.1 | 0.1 | 0.1 |
| BLM: DeBERTaV3-large | | | | | | | | |
| learning rate (lr) | 5e-4 | 4e-4 | 8e-4 | 7e-4 | 9e-5 | 1e-3 | 1.2e-4 | 4e-4 |
| batch size | 16 | 16 | 16 | 16 | 16 | 16 | 16 | 16 |
| lr scheduler | cosine | cosine | cosine | cosine | cosine | cosine | cosine | cosine |
| epochs | 30 | 24 | 50 | 40 | 8 | 50 | 10 | 8 |
| max length | 64 | 128 | 320 | 256 | 320 | 320 | 512 | 256 |
| weight decay | 0.1 | 0.1 | 0.1 | 0.1 | 0.1 | 0.1 | 0.1 | 0.1 |

## C    INFERENCE TIME COMPARISONS

We compare the inference time of our approach with the LST method and the baseline using only the BLM. For LoRA, the inference time is identical to that of the BLM because, in evaluation mode,

Table 9: Hyper parameters of decoder-only model experiments

| / | CoLA | SST2 | MRPC | STSB | QQP | RTE | QNLI | MNLI |
|---|---|---|---|---|---|---|---|---|
| BLM: OPT-1.3b | | | | | | | | |
| learning rate (lr) | 2e-4 | 2e-4 | 2e-4 | 2e-4 | 2e-4 | 2e-4 | 2e-4 | 2e-4 |
| batch size | 16 | 16 | 16 | 16 | 16 | 16 | 16 | 16 |
| lr scheduler | cosine | cosine | cosine | cosine | cosine | cosine | cosine | cosine |
| epochs | 30 | 24 | 50 | 40 | 8 | 50 | 10 | 8 |
| max length | 64 | 128 | 320 | 256 | 320 | 320 | 320 | 256 |
| weight decay | 0.1 | 0.1 | 0.1 | 0.1 | 0.1 | 0.1 | 0.1 | 0.1 |
| BLM: OPT-2.7b | | | | | | | | |
| learning rate (lr) | 2e-4 | 2e-4 | 2e-4 | 2e-4 | 2e-4 | 2e-4 | 2e-4 | 2e-4 |
| batch size | 8 | 8 | 8 | 8 | 8 | 8 | 8 | 8 |
| lr scheduler | cosine | cosine | cosine | cosine | cosine | cosine | cosine | cosine |
| epochs | 30 | 24 | 50 | 40 | 8 | 50 | 10 | 8 |
| max length | 64 | 128 | 320 | 256 | 320 | 320 | 512 | 256 |
| weight decay | 0.1 | 0.1 | 0.1 | 0.1 | 0.1 | 0.1 | 0.1 | 0.1 |

Table 10: Hyper parameters of multi-label experiments

| / | Semeval2014-task1 | Semeval2018-task1 |
|---|---|---|
| learning rate (lr) | 8e-4 | 8e-4 |
| batch size | 16 | 16 |
| lr scheduler | cosine | cosine |
| epochs | 40 | 32 |
| max length | 328 | 128 |
| weight decay | 0.1 | 0.1 |

the LoRA modules are integrated into the weight matrix. The inference times for a single run with a sequence length of 512 and a batch size of 4 on OPT-1.3b, OPT-2.7b, and OPT-6.7b are presented in Table 11.

The experimental results reveal some inference time limitations in side-tuning methods. However, our method requires fewer parameters and, consequently, fewer floating-point operations compared to LST, resulting in slightly shorter inference times.

Table 11: Inference time comparisons on OPT series models. We conducted our experiments on a single NVIDIA RTX A6000 GPU with 48 GB of memory. The inference times, measured in seconds, are reported.

| / | BLM | LST | S.T. (Ours) |
|---|---|---|---|
| OPT-1.3b | $0.313(\times 1.00)$ | $0.391(\times 1.25)$ | $0.387(\times 1.24)$ |
| OPT-2.7b | $0.594(\times 1.00)$ | $0.700(\times 1.17)$ | $0.684(\times 1.15)$ |
| OPT-6.7b | $1.442(\times 1.00)$ | $1.673(\times 1.16)$ | $1.601(\times 1.11)$ |

