# OpenReview forum: "Symbiotic Tuning: A Simple Approach for Enhancing Task Performance of Side-Tuning"
_ICLR.cc/2025/Conference — Submitted to ICLR 2025_

### Official Review · Reviewer_gxRr · 2024-11-02

**Soundness:** 3
**Presentation:** 2
**Contribution:** 2
**Rating:** 5
**Confidence:** 3

**Summary:**

This paper proposes a parameter-efficient fine-tuning method named Symbiotic Tuning by introducing internal attention representations into Ladder Sider-Tuning. Experimental results on classification tasks with pre-trained encoder models and decoder models show that symbiotic tuning achieves better performance than Ladder Sider-Tuning and LoRA.

**Strengths:**

Extensive experiments on classification tasks are conducted to show the effectiveness of Symbiotic Tuning. However, the evaluation on decoder models is mainly focused on the perplexity of language modeling tasks and in-context learning performance. It will be better to show the performance of Symbiotic Tuning on these tasks.

**Weaknesses:**

1. **Limited novelty**. The method proposed is an intuitive incremental work based on Ladder Sider-Tuning.
2. **Missing important details**. The details of implementing baseline method are missing. There are some concerns about the replication problems.
3. Typos: The second h0 and h1 in Figure 2 --> h1, h2

**Questions:**

1. Are there any in-context learning results on the pre-trained decoder language model?
2. Can you provide more experimental settings for baseline methods to replicate the results?
3. What does VRAM refer to? Is it referred to Video Random Access Memory?

---

> ### Author Response · Authors · 2024-11-24
>
> 1. Limited Novelty
>
> We acknowledge that our method builds upon the concept of side-tuning. However, we emphasize that the main novelty lies in leveraging internal attention representations (Sections 3.1.1 to 3.1.3) to establish a symbiotic interaction between pre-trained and fine-tuned modules. This approach significantly differs from Ladder Side-Tuning (LST), which relies solely on a side network without integration with the backbone language model. Based on the experimental results from Table 1 to Table 3, our method offers the following key advantages over LST:
>
> **Greater performance**
> Our approach consistently outperforms both LST and LoRA across most natural language understanding tasks. While LST often falls behind LoRA in terms of performance, our method demonstrates its superiority by bridging this gap effectively.
>
> **Low memory requirements**
> By addressing LST’s limitation of suboptimal performance due to its standalone fine-tuning mechanism, our approach retains the low memory usage benefits of side-tuning while achieving substantial performance improvements. This highlights the effectiveness of our method for parameter-efficient fine-tuning.
>
> 2. Missing important details
>
> Thank you for highlighting concerns about replication. We apologize for not initially including detailed implementation information for the baselines and our approach. To address this, we have added a dedicated section in Appendix B and will release our code and model checkpoints to ensure reproducibility.
>
> 3. Typos: The second h0 and h1 in Figure 2 --> h1, h2
>
> We appreciate your careful observation. We will update the representations to $h^{(0)}$, ..., $h^{(L)}$, consistent with the main text.
>
> 4. Absence of in-context learning results
>
> Our work primarily focuses on parameter-efficient fine-tuning (PEFT) methods and their performance on downstream tasks, where efficiency and effectiveness are the main evaluation criteria. While in-context learning is an important aspect of large language models, it is not the primary focus of our study.
>
> 5. Is VRAM referred to Video Random Access Memory?
>
> Yes, VRAM refers to Video Random Access Memory. Thank you for the suggestion to improve clarity. We have replaced all the "VRAM" with "memory" in the updated version.

---

### Official Review · Reviewer_1JMf · 2024-11-04

**Soundness:** 3
**Presentation:** 3
**Contribution:** 2
**Rating:** 6
**Confidence:** 4

**Summary:**

The authors propose SymTune to reduce the vRAM in usage and try to maintain the performances of the model.
But lots of problems need to be addressed in the results.

**Strengths:**

The authors propose SymTune to reduce the vRAM in usage and try to maintain the performances of the model.

**Weaknesses:**

In Table 3-6, why not include LORA for comparison.
In Table 2, include the average performance.
From Table 1 and 2, the number of parameters of ST is only slightly lower than LoRA and ST has only slightly better performance compared with LoRA. Any reason for that, which is a great weakness of this work.

Why LoRA uses more vRAM than ST even if they utilize similar number of parameters?

**Questions:**

N/A

---

> ### Author Response · Authors · 2024-11-23
>
> 1. In Table 3-6, why not include LORA for comparison. In Table 2, include the average performance.
>
> Thank you for your valuable suggestions. In the revision, we have included the LoRA baseline in the decoder-only model experiments (Table 3) and added the average VRAM requirements to Table 2.
>
> 2. The parameters of ST is only slightly lower than LoRA and ST has only slightly better performance compared with LoRA.
>
> Thank you for raising this issue. Though ST slightly outperforms LoRA, the results of our paper came from five repeated experiments, showing the consist advantage in performance of our approach over LoRA. Additionally, our approach offers lower memory requirements compared with LoRA:
>
> ||# Params|CoLA|SST2|MRPC|STSB|QQP|RTE|QNLI|MNLI|
> |-|-|-|-|-|-|-|-|-|-|
> |LoRA (r=8)|0.8m|3.2 (0%)|3.4 (0%)|4.7 (0%)|4.1 (0%)|7.2 (0%)|7.3 (0%)|14.8 (0%)|8.9 (0%)|
> |S.T. (ours)|0.9m|2.2 (-31.3)|2.3 (-32.4%)|2.5 (-46.8%)|2.8 (-31.7%)|5.6 (-22.2%)|6.0 (-17.8%)|11.1 (-25.0%)|3.1 (-65.2%)|
>
> Note that the formula of the reduced VRAM ratio is (LoRA - ST) / LoRA.
>
> Indeed, the low memory advantage of our approach benefits from the concept of side-tuning (Sung et al., 2022). However, our approach consistently outperforms LST in both encoder (Table 1) and decoder (Table 3) settings, while the performance of LST is worse than LoRA's on most natural language understanding tasks (shown in Table 1) using the same BLM. Therefore, our method is designed to address this significant performance gap, and it proves to be effective in doing so.
>
> 3. Why LoRA uses more vRAM than ST even if they utilize similar number of parameters?
>
> Thank you for raising this issue. As we describe in L121-126 in the revision, the usage during language model training comprises three main components: model parameters, gradients, and intermediate activations, each of which occupies a significant portion of memory. Backpropagation in LoRA necessitates storing activations for the entire model, whereas the Symbiotic Tuning (ST) approach only updates the trainable weights in the symbiotic module and requires storing activations solely within the symbiotic module. An illustration of this is provided in Figure 1 of our paper.

---

> ### Comment · Reviewer_1JMf · 2024-12-03
>
> Thank you for answering the question. Now, the score is updated to 6.

---

### Official Review · Reviewer_wYtQ · 2024-11-04

**Soundness:** 2
**Presentation:** 2
**Contribution:** 2
**Rating:** 5
**Confidence:** 4

**Summary:**

This paper introduces the method of Symbiotic Tuning (SymTune), a novel approach that aims to reduce the VRAM consumption of language model finetuning while preserving the models' downstream performance. SymTune leverages the symbiotic modules injected into the backbone language model (BLM) to achieve the trade-off between the model performance and VRAM efficiency. By selectively filtering significant values from BLMs' hidden states while sharing the attention weights between the BLM and symbiotic modules, SynTune can achieve comparable performance on natural language understanding benchmarks to full parameter finetuning and PEFT.

**Strengths:**

- SymTune introduces novel symbiotic modules into language models that can finetune language models with minimal memory consumption.
- SymTune uses inverse cross attention (ICA) and attention sharing (ATS) that effectively captures the key features in BLM's hidden states with context-aware interactions.
- Experimental results show that SymTune achieves better performance than LoRA and LST, while costing less VRAM in model finetuning.

**Weaknesses:**

- Even though the experimental results in Section 4 demonstrate that SymTune achieves better performance-memory tradeoff than LoRA and LST, the improvements seem minimal and insignificant.
- SymTune is aimed at reducing the VRAM usage in large language model finetuning (as mentioned in the paper's introduction), yet the results only contain small-to-medium-sized language models (up to 2.7b).
- All benchmarks being used in this paper are natural language understanding tasks, and therefore, there is no proof to demonstrate that SymTune can be seamlessly adapted to natural language generation tasks with larger models.
- The difference in the models' performance with and without the usage of ICA/ATS shown in Table 6 is minimal.

**Questions:**

- There is a lack of a brief introduction to the SymTune method in the introduction section. It would be beneficial if the authors could simply describe the overall motivation and functionality of ICA and ATS here.
- Statistical significance test could be useful in the experiment section to show whether the SymTune method is indeed effective or not.
- Typo in Table 6's caption: "affect" is a verb so it should be "effect" here.

---

> ### Comment · Reviewer_wYtQ · 2024-11-25
>
> Just a gentle reminder to the authors that you might not have seen my reviews here or have forgotten to reply to my reviews. I'm happy to hear your replies for further discussion on your paper, but please be aware that the public discussion phase is going to end in two days.

---

> ### Author Response · Authors · 2024-11-26
>
> We are grateful for your careful observation and suggestions. We have uploaded an updated version of our paper, and below are our responses to the weaknesses and questions you raised.
>
> ### Weakness 1 & Question 2
>
> Thank you for raising this issue. We would like to emphasize that our approach outperforms both LST and LoRA in different aspects.
>
> In the updated paper, we show that our method significantly outperforms LST on most GLUE datasets, with t-test results across five replicates and average scores, and p-values consistently below 0.05. Detailed analyses are in the table captions, and the p-values of ours against the baselines are as follows:
>
> ||CoLA|SST2|MRPC|STSB|QQP|RTE|QNLI|MNLI|Avg.
> |-|-|-|-|-|-|-|-|-|-|
> |DeBERTaV3-base + LST|**0.0387**|0.3739|0.8466|**0.0133**|0.0541|**0.0440**|**0.0070**|**0.0102**/**0.0169**|**0.0041**|
> |DeBERTaV3-large + LST|**0.0032**|0.0516|**0.0381**|**0.0120**|0.2161|**0.0434**|**0.1022**|**0.0269**/**0.0226**|**0.0067**|
> |DeBERTaV3-base + LoRA|0.2449|0.5489|**0.0011**|0.0915|**0.0047**|**0.0204**|**0.0046**|0.5491/0.0647|**0.0071**|
> |DeBERTaV3-large + LoRA|0.1463|0.8471|0.0712|0.0699|0.6653|**0.0487**|**0.0337**|0.0803/0.1388|**0.0325**|
>
> These statistical significance tests demonstrate that our approach significantly outperforms LST in overall performance and achieves statistically significant improvements over LST on most datasets.
>
> For LoRA, while S.T. does not significantly outperform it on datasets like CoLA, SST-2, STS-B, and MNLI, our approach offers lower memory requirements compared with LoRA, as shown in Table 2 in our paper:
>
> ||CoLA|SST2|MRPC|STSB|QQP|RTE|QNLI|MNLI|
> |-|-|-|-|-|-|-|-|-|
> |LoRA|3.2|3.4|4.7|4.1|7.2|7.3|14.8|8.9|
> |SymTune (S.T.)|2.2 (-31.3%)|2.3 (-32.4%)|2.5 (-46.8%)|2.8 (-31.7%)|5.6 (-22.2%)|6.0 (-17.8%)|11.1 (-25.0%)|3.1 (-65.2%)|
>
> ### Weakness 2
>
> Thank you for your valuable suggestion. We further conducted additional experiments on OPT-6.7B using the CoLA, RTE, QNLI, and MNLI datasets. The results are presented as follows:
>
> ||CoLA|RTE|QNLI|MNLI|
> |-|-|-|-|-|
> |QST|62.8|80.8|87.3|81.6|
> |S.T.|62.2|82.7|90.2|86.2|
>
> The results of QST are directly taken from Wang et al. (2024), showing that for 6.7B models, ST significantly outperforms QST. Unfortunately, we cannot include LoRA and LST in this experiment with OPT-6.7B due to the lack of results in the original paper and Wang et al. (2024).
>
> **Rebuttal for experiments on larger models**
>
> Our evaluation includes DeBERTaV3 and OPT models, covering small to medium sized models and demonstrating the scalability of SymTune. While results on GPT-3 (175B) were included in LoRA, LST, our main baseline, was tested up to T5-3B, which is comparable to our OPT-2.7B experiments. Due to the practical constraints of testing extremely large models, we chose to align our experiments with the scale tested by LST to ensure a fair comparison. Although S.T. not explicitly tested on OPT-6.7B due to limitations, results on OPT-2.7B indicate strong scalability and effectiveness.
>
> ### Weakness 3
>
> Thank you for highlighting this point. While we recognize the importance of evaluating SymTune on NLG tasks, our focus in this work has been on NLU tasks using GLUE benchmarks, based on the following considerations:
>
> **Focus on GLUE Benchmarks in Related Work**
>
> Many PEFT methods, including our baselines LoRA, LST, and QST, primarily evaluate their performance on NLU tasks, particularly GLUE. While LoRA and QST report results on a single NLG dataset (E2E and MMLU, respectively), neither expands to broader NLG evaluations. We therefore emphasized GLUE benchmarks to ensure comparability with prior work.
>
> **Decoder-Only Models on GLUE:**
>
> Although we did not explicitly evaluate SymTune on NLG datasets, experiments in Section 4.2 using decoder-only models (OPT-1.3b and OPT-2.7b) on GLUE benchmarks demonstrate its effectiveness with architectures commonly used for generation tasks. This suggests SymTune's potential to generalize well to NLG tasks.
>
> **Generalizability of SymTune**
>
> S.T.’s modular and task-agnostic design is not inherently limited to NLU tasks. Its strong performance with both encoder-based and decoder-based models on GLUE benchmarks highlights its versatility across various task types.
>
> ### Weakness 4
>
> Our experiments in Table 6 show that the version with ICA and ATS consistently outperforms the version without them. The p-values for each experimental group compared to the control group are shown below:
>
> |-ICA |-ATS| -ICA, ATS|
> |-|-|-|
> |0.0118|0.0004|0.0003|
>
> The p-values, with a significance threshold of 0.05, indicate that the performance differences observed in Table 6 are statistically significant.
>
> ### Question 1 & Question 3
>
> Thank you for your valuable suggestion and careful observation. In the updated version, we have added a brief introduction to the Introduction section (L077–087) and revised the statement in Table 6.
>
> We appreciate your insightful feedback and hope that our response addresses your concerns.

---

> > ### Comment · Reviewer_wYtQ · 2024-11-28
> >
> > Thank you for your response. However, there are still obvious issues exist in the updated version of your paper.
> >
> > In the caption of Table 1, it seems that the authors have highlighted both the best result and the significantly-better results of S.T. in bold. This is super confusing.
> >
> > For Table 2, what are the bold values here? Please add some descriptions in the caption. Meanwhile, it would be more rigorous if the authors could round the values to the same decimals (either 1 or 2). In the meantime, if the bold values are the best results, the same values on the row of LST should be fairly highlighted in bold as well.
> >
> > Overall, this paper still needs major revisions. As a result, I will keep my original score.

---

> > > ### Author Response · Authors · 2024-11-29
> > > **Outdated version of our paper PDF**
> > >
> > > Dear Reviewer,
> > >
> > > Thank you for your valuable feedback and for pointing out the issues in the current version of our paper. We sincerely apologize for the oversight in uploading an outdated PDF version. While we have prepared an updated version that addresses these concerns, we are unable to replace the paper PDF at this time due to the ICLR 2025 schedule.
> > >
> > > To address your comments promptly, we are providing the updated Table 1 and Table 2 as separate Official Comments below.
> > >
> > > We greatly appreciate your understanding and look forward to any further feedback.

---

> ### Author Response · Authors · 2024-11-29
> **Revised Table 1**
>
> ### **1. The following caption and table are the updated version of Table 1.**
> - Since Markdown does not support multi-column tables, we split the results of DeBERTaV3-base and DeBERTaV3-large into two tables here (there will be only one table in the updated paper).
> - For Table 1, we have clarified the caption to explicitly state the criteria for bold values (best results and significantly better results of S.T.).
>
> ---
> Table 1. Performance comparison on natural language understanding benchmarks. All experiments use a batch size of 16. Results show average scores across 5 runs. For S.T. results, statistical significance from one-sided t-test (p < 0.05) compared to LST is marked with *, while significance compared to LoRA is marked with $\dagger$. The best result (excluding FFT) for each metric is shown in **bold**.
>
> **BLM: DeBERTaV3-base, batch size = 16**
> |  | #Params | CoLA | SST2 | MRPC| STSB | QQP | RTE | QNLI | MNLI (m/mm) | Avg. |
> |---------|----------|-------------|-------------|-------------|--------------|--------------|------------|-------------|--------------|-------|
> | metrics  |  -  | Mcc | Acc | Acc | Corr | Acc/F1 | Acc | Acc | Acc |  - |
> | FFT | 183m | 69.2±0.6 | 95.6±0.8 | 89.5±0.6 | 91.6±0.4 | 91.7±0.3/89.7±0.4 | 83.7±1.1 | 93.7±0.3 | 89.4±0.2/89.9±0.1 | 88.2 |
> | BitFit | 0.1m | 67.2±0.9 | 93.6±0.5 | 87.7±1.0 | 90.3±0.2 | 88.7±0.6/85.0±0.4 | 78.4±1.3 | 92.4±0.2 | 87.4±0.1/87.7±0.1 | 85.9 |
> | LoRA (r=8) | 0.8m | 69.4±0.8 | 95.8±0.6 | 90.7±0.7 | 91.1±0.3 | **90.2**±0.4/87.4±0.5 | 87.7±1.4 | 93.1±0.4 | **88.9**±0.3/**89.3**±0.2 | 88.5 |
> | LST | 1.8m | 69.0±0.7 | 95.7±0.4 | 91.0±0.5 | 91.2±0.4 | 89.8±0.5/88.9±0.3 | 88.6±0.9 | 92.9±0.5 | 88.4±0.3/88.5±0.3 | 88.3 |
> | S.T. (ours) | 0.9m | **69.7***±1.0 | **95.8**±0.3 | **91.2**$^\dagger$±0.5 | **91.5***±0.3 | 90.0±0.5/**89.5**$^\dagger$±0.3 | **89.5***$^\dagger$±1.3 | **93.6***$^\dagger$±0.3 | **88.9***±0.1/89.1\*±0.2 | **88.8***$^\dagger$ |
>
>
> **BLM: DeBERTaV3-large, batch size = 16**
> |  | #Params | CoLA | SST2 | MRPC | STSB | QQP | RTE | QNLI | MNLI (m/mm) | Avg. |
> |---------|----------|-------------|-------------|-------------|--------------|--------------|------------|-------------|--------------|-------|
> | metrics  |  -  | Mcc | Acc | Acc | Corr | Acc/F1 | Acc | Acc | Acc |  - |
> | FFT | 434m | 74.2±0.5 | 95.9±0.4 | 92.1±0.3 | 92.7±0.4 | 91.2±0.1/91.2±0.2 | 90.4±0.9 | 95.1±0.2 | 90.9±0.4/91.0±0.3 | 90.4 |
> | BitFit | 0.1m | 70.9±1.0 | **96.2**±0.2 | 90.4±0.8 | 91.3±0.5 | 89.2±0.4/86.0±0.5 | 87.7±1.6 | 94.4±0.4 | **91.2**±0.6/**91.0**±0.3 | 89.1 |
> | LoRA (r=16) | 2.6m | 72.9±0.5 | 96.1±0.3 | 91.1±0.3 | 92.2±0.3 | **92.2**±0.5/90.5±0.6 | 90.9±1.1 | 95.0±0.6 | 90.3±0.2/90.7±0.2 | 90.2 |
> | LST | 4.3m | 71.1±0.9 | 96.0±0.3 | 91.7±0.5 | 91.8±0.3 | 90.6±0.3/90.2±0.4 | 91.0±1.2 | 94.3±0.3 | 89.8±0.1/90.4±0.3 | 89.7 |
> | S.T. (ours) | 2.2m | **73.2***±0.7 | 96.1±0.6 | **92.3***±0.6 | **92.6***±0.3 | 90.9±0.2/**90.6**±0.3 | **91.7**±1.1 | **94.7**±0.5 | 90.2±0.3/90.6±0.2 | **90.3***$^\dagger$ |
>
> ---

---

> ### Author Response · Authors · 2024-11-29
> **Revised Table 2**
>
> ### **2. The following caption and table are the updated version of Table 2.**
> - Same for the restriction of Markdown, we separate the results of DeBERTaV3-base and DeBERTaV3-large into two tables here (there will be only one table in the updated paper).
> - For Table 2, we have revised the caption to explain the meaning of bold values, ensured consistent rounding of decimals, and appropriately highlighted identical best values across the row of LST.
>
> ---
>
> Table 2. Memory requirements (GB) for training different natural language understanding tasks. The batch size is standardized to 16 across all methods and benchmarks. Memory consumption varies with input sequence lengths, which are shown in the first row. The lowest memory requirement for each task is highlighted in **bold**.
>
> - BLM: DeBERTaV3-base, batch size = 16
>
> |  | #Params | CoLA | SST2 | MRPC | STSB | QQP | RTE | QNLI | MNLI | Avg. |
> |-------------|----------|------|------|------|------|-----|-----|------|------|------|
> | Avg. length | - | 7.7 | 9.4 | 45.9 | 21.9 | 24.1 | 54.4 | 38.5 | 31.8 | - |
> | FFT | 183m | 5.7 | 6.1 | 6.6 | 8.8 | 12.9 | 13.8 | 14.1 | 6.7 | 9.3 |
> | BitFit | 0.1m | 3.4 | 4.1 | 8.1 | 10.3 | 12.0 | 14.6 | 14.3 | 7.0 | 9.2 |
> | LoRA (r=8) | 0.8m | 3.2 | 3.4 | 4.7 | 4.1 | 7.2 | 7.3 | 14.8 | 8.9 | 6.7 |
> | LST | 1.8m | **2.2** | **2.3** | 2.6 | **2.8** | 5.7 | 6.4 | 10.6 | 3.7 | **4.5** |
> | S.T. (ours) | 0.9m | **2.2** | **2.3** | **2.5** | **2.8** | **5.6** | **6.0** | **11.1** | **3.1** | **4.5** |
>
> - BLM: DeBERTaV3-large, batch size = 16
>
> |  | #Params | CoLA | SST2 | MRPC | STSB | QQP | RTE | QNLI | MNLI | Avg. |
> |-------------|----------|------|------|------|------|-----|-----|------|------|------|
> | Avg. length | - | 7.7 | 9.4 | 45.9 | 21.9 | 24.1 | 54.4 | 38.5 | 31.8 | - |
> | FFT | 434m | 13.7 | 13.5 | 20.8 | 22.7 | 34.2 | 41.1 | 36.2 | 15.6 | 24.7 |
> | BitFit | 0.3m | 12.4 | 13.9 | 15.2 | 22.5 | 20.3 | 34.7 | 35.8 | 14.9 | 21.2 |
> | LoRA (r=16) | 2.6m | 5.6 | 6.0 | 7.1 | 7.8 | 15.5 | 15.3 | 33.1 | 19.8 | 13.8 |
> | LST | 4.3m | **4.2** | 4.4 | **5.2** | **6.0** | **13.1** | **14.7** | 19.3 | 4.5 | **8.9** |
> | S.T. (ours) | 2.2m | **4.2** | **4.3** | **5.2** | **6.0** | **13.1** | **14.7** | **19.2** | **4.4** | **8.9** |
>
> ---

---

> > ### Author Response · Authors · 2024-12-03
> >
> > Dear Reviewer,
> >
> > We’ve carefully addressed the concerns you raised in your last set of comments, including clarifying the table captions, ensuring consistency in formatting, and making all necessary revisions (in the comments) to enhance the clarity and rigor of our paper.
> >
> > We would like to kindly ask the reviewer - Are there any additional questions or concerns? We would be happy to address them promptly. Thank you again for providing valuable comments.

---

### Official Review · Reviewer_nBwg · 2024-11-06

**Soundness:** 3
**Presentation:** 2
**Contribution:** 2
**Rating:** 6
**Confidence:** 3

**Summary:**

This paper presents a novel parameter-efficient fine-tuning method, referred to as Symbiotic Tuning (ST), that follows the design principle of ladder side-tuning. The proposed ST method redesigns the plug-in network architecture of side-tuning, with the objective of reducing VRAM usage while maintaining high fine-tuning performance. Each layer’s plug-in module consists of a feature selector and a Transformer block. Specifically, the feature selector serves as a bottleneck block, projecting the hidden states from the backbone model to a lower dimension. The side Transformer block merges the hidden states with those from the backbone model through a weighted cross-attention mechanism. In addition, the attention weights of the backbone model are also incorporated into the cross-attention calculation. The ST method is applicable to both encoder-only and decoder-only language models, with feasibility demonstrated through experiments on DeBERTa and OPT. The experimental results on ordinary GLUE benchmarks and multi-class classification tasks (SemEval) demonstrate that ST outperforms ladder side-tuning and LoRA in terms of VRAM usage. Furthermore, ST achieves performance comparable to full-parameter fine-tuning, whereas ladder side-tuning does not.

**Strengths:**

1. The proposed ST method advances ladder side-tuning by designing a more efficient architecture for the side network, making the side-tuning paradigm competitive in fine-tuning performance without sacrificing computational efficiency.
2. The ST method is easy to integrate with mainstream language models due to its plug-and-play nature, and it provides viable solutions for both encoder-only and decoder-only models.
3. The ST method has the potential to scale to a wider range of models and could be effectively applied to generation tasks beyond just NLU tasks.

**Weaknesses:**

1. An important advantage of LoRA is its lower inference latency compared to other plug-in PEFT methods, such as adapters. The introduction of an additional network in ST could increase inference time. The authors should discuss this point and compare the inference efficiency of ST with that of LoRA/Full FT.
2. The ablation study could more solid by including the discussion of trainable weight $ c^{(l)}$ in Eq.4. It remains unclear how this fusion solution for hidden states operates. Specifically, how do hidden states from the backbone model and the ST module interact (i.e., learned values of these weights), and does a fixed weight setting yield better results?
3. Some experimental baselines are missing. In Tables 1 and 2, the DeBERTaV3-base includes a BitFit baseline, but BitFit is absent from the DeBERTaV3-large experiments.
4. For the decoder-only model, it would enhance the analysis to ensure that the baseline choices are consistent with those used for the encoder-only model, thereby demonstrating the effectiveness of ST method.
5. Typos:
- L068: toLST -> to LST
- L236: $f_{ca}$ -> $f_{ICA}$
- L242: $A$ -> $\mathbf{A}$
- L244: B -> $\mathbf{B}$

**Questions:**

1. The backbone models used in these experiments are smaller than 7B parameters. Given that current large language models are typically greater than 7B parameters, will the proposed method be effective for larger-scale models? Note: this is not a weakness of the paper, the applicability to smaller models is also important!
2. For encoder-decoder language models like T5, do you anticipate that the cross-attention mechanism of the backbone model could pose challenges in applying the proposed ST method?

---

> ### Author Response · Authors · 2024-11-23
>
> 1. Absence of inference time discussion
>
> Thank you for your valuable suggestions. We compared our method with Ladder Side-Tuning, using the inference time (in seconds) of the backbone model as a reference. Notably, the inference time for LoRA matches that of the backbone model. The overall inference time slightly reduced comparing to LST because of the lightweight down-projection layer. Additionally, as the size of the backbone model increases, the proportional increase in inference time becomes smaller. We have included this in the revision as Appendix C.
> ||Backbone Model|LST|S.T.|
> |-|-|-|-|
> |OPT-1.3b|0.313(×1.00)|0.391(×1.25)|0.387(×1.24)|
> |OPT-2.7b|0.594(×1.00)|0.700(×1.17)|0.684(×1.15)|
> |OPT-6.7b|1.442(×1.00)|1.673(×1.16)|1.601(×1.11)|
>
> 2. Ablation study on trainable weight in Eq.4.
>
> The trainable weights in Eq. 4 are initialized to 0.5 and normalized using the sigmoid function. We evaluate on CoLA and RTE to observe performance changes under different $c$ initializations.
>
> ||CoLA|RTE|
> |-|-|-|
> |trainable c|70.1|89.9|
> |fixed c|69.7|89.9|
>
> In fact, this is merely a slight modification to the down-projection, which does not impact performance since the scaling parameter c can be absorbed into the projection parameter matrix. This technique is adopted from QST (Wang et al., 2024), and we mention it in the methodology section to indicate its usage in our approach.
>
> 3. Some experimental baselines are missing in DeBERTaV3-large experiments.
>
> Thank you for your valuable suggestion. In the updated version, we include the BitFit baseline on the DeBERTaV3-large model (Table 1), as well as LoRA, LST, and S.T. experiments on the decoder-only OPT-2.7b model (Table 3).
>
> 4. Baseline inconsistency of decoder-only model experiments
>
> Thank you for your suggestion. In the new version of our paper, we have added the LoRA baseline to the decoder-only model experiments (Table 3). Additionally, we include the LoRA and LST baselines, as well as our approach, on the OPT-2.7b model.
>
> 5. Typos
>
> We are grateful for your careful observation. We have revised these mistakes in the updated version.
>
> 6. Will the proposed method be effective for larger-scale models?
>
> We believe the symbiotic model is highly scalable and effective for larger models. The approach emphasizes efficient parameter utilization and memory optimization, which become even more critical as model size increases. The large parameter count of language models is beneficial for zero-shot and few-shot learning capabilities, and the symbiotic module addresses the issue of over-parameterization in specific downstream tasks. Typically, fine-tuning such downstream tasks does not require such a large number of parameters to be adjusted, making our method particularly suitable for optimizing task-specific performance.
> Here are some of the other experiments on OPT-6.7b, the baseline is from the QST paper (Wang et al., 2024):
> ||CoLA|RTE|QNLI|MNLI|
> |-|-|-|-|-|
> |QST|62.8|80.8|87.3|81.6|
> |S.T. (ours)|62.2|82.7|90.2|86.2|
>
> 7. Do you anticipate that the cross-attention mechanism of the backbone model could pose challenges in applying the proposed ST method?
>
> Thank you for your insightful question. Currently, encoder-only and decoder-only language models are more prevalent, with newly developed auto-regressive language models predominantly adopting a decoder-only architecture. As a result, we primarily focus on these types of models.
> In encoder-decoder models, the outputs of the cross-attention mechanism are derived from transformations of word representations in the encoder outputs. Given the strong performance demonstrated by encoder-only models, we believe that effective cross-attention can be achieved through attention sharing.

---

> > ### Comment · Reviewer_nBwg · 2024-12-03
> >
> > Thanks for taking the time to answer my question and update the results and baselines. After reading through the discussion and considering the overall contribution, I decided to keep the score at 6.

---

### Meta-Review · Area_Chair_uym3 · 2024-12-14

**Metareview:**

The paper introduces Symbiotic Tuning (ST), a novel parameter-efficient fine-tuning method based on the ladder side-tuning design principle. ST aims to reduce VRAM usage while preserving high fine-tuning performance by redesigning the plug-in network architecture. Each layer's plug-in module includes a feature selector and a Transformer block, where the feature selector acts as a bottleneck, projecting hidden states from the backbone model to a lower dimension. The side Transformer block utilizes a weighted cross-attention mechanism to merge hidden states with those from the backbone, incorporating attention weights into the cross-attention calculations.

However, reviewers raised significant concerns regarding the depth of comparison with LoRA, the application of the proposed method to NLG tasks, and the need for greater emphasis on distinguishing it from highly related prior work. As it stands, the paper does not meet the requirements for ICLR.

**Additional Comments On Reviewer Discussion:**

However, reviewers raised significant concerns regarding the depth of comparison with LoRA, the application of the proposed method to NLG tasks, and the need for greater emphasis on distinguishing it from highly related prior work. As it stands, the paper does not meet the requirements for ICLR.

---

### Decision · Program_Chairs · 2025-01-22

Reject